# Development of Novel Colorful Electrorheological Fluids

**DOI:** 10.3390/nano12183113

**Published:** 2022-09-08

**Authors:** Suk Jekal, Jiwon Kim, Qi Lu, Dong-Hyun Kim, Jungchul Noh, Ha-Yeong Kim, Min-Jeong Kim, Min-Sang Kim, Won-Chun Oh, Hyoung-Jin Choi, Chang-Min Yoon

**Affiliations:** 1Department of Chemical and Biological Engineering, Hanbat National University, Yuseong-gu, Daejeon 34158, Korea; 2Department of Polymer Science and Engineering, Inha University, Michuhol-gu, Incheon 22212, Korea; 3Program of Environmental and Polymer Engineering, Inha University, Michuhol-gu, Incheon 22212, Korea; 4McKetta Department of Chemical Engineering and Texas Material Institute, The University of Texas at Austin, Austin, TX 78712, USA; 5Department of Advanced Materials Science and Engineering, Hanseo University, Seosan-si 31962, Korea

**Keywords:** electrorheological fluids, colorful, pearlescent, TiO_2_ coating, light interference

## Abstract

Herein, the electrorheological (ER) performances of ER fluids were correlated with their colors to allow for the visual selection of the appropriate fluid for a specific application using naked eyes. A series of TiO_2_-coated synthetic mica materials colored white, yellow, red, violet, blue, and green (referred to as color mica/TiO_2_ materials) were fabricated via a facile sol–gel method. The colors were controlled by varying the thickness of the TiO_2_ coating layer, as the coatings with different thicknesses exhibited different light interference effects. The synthesized color mica/TiO_2_ materials were mixed with silicone oil to prepare colored ER fluids. The ER performances of the fluids decreased with increasing thickness of the TiO_2_ layer in the order of white, yellow, red, violet, blue, and green materials. The ER performance of differently colored ER fluids was also affected by the electrical conductivity, dispersion stability, and concentrations of Na^+^ and Ca^2+^ ions. This pioneering study may provide a practical strategy for developing new ER fluid systems in future.

## 1. Introduction

Electrorheological (ER) fluids, which are well-dispersed suspensions of polarizable materials in insulating media, such as silicone oils, have received widespread attention owing to their various advantages, such as rapid response, low power consumption, and reversible phase transition [1,2,3,4,5]. When an external electric field (*E*-field) is applied, ER fluids exhibit a sudden and rapid change from liquids to solid-like states within a few milliseconds [6,7,8,9,10]. ER fluids can therefore be used in various industrial applications which require fast response times, including dampers, clutch systems, engine mounts, and haptic devices [11,12,13]. ER activities can be improved by controlling the mechanical and chemical properties of dispersing materials [14,15,16]. Additionally, the polarizability of the materials plays a pivotal role in ER performance [17,18]: in general, materials with high polarizability exhibit high ER efficiencies [19]. Various polarizable materials, including metallic, polymeric, organic, and inorganic materials, have been used as ER materials [20,21,22].

Among various inorganic and metal oxides, TiO_2_ is widely used as a dispersing material for ER fluids owing to its advantages, such as availability for mass production and simple fabrication process via the sol–gel method [23,24,25]. TiO_2_ and its derived materials also exhibit numerous chemical advantages, such as stability, resistance to corrosion, and photocatalytic ability [26,27,28,29,30]. Previous ER studies have developed numerous TiO_2_-derived materials, such as polymer-coated TiO_2_, shape-controlled TiO_2_, and alkaline earth metal-doped TiO_2_ [31,32]. Furthermore, TiO_2_ materials can be incorporated or coated onto a SiO_2_ template to fabricate multilayered materials to achieve core/shell structures, porous characteristics, and precursors for hollow-type materials [33,34,35]. Silicon-containing materials, such as SiO_2_, SiO_3_, and other silicon-containing oxides, are widely utilized as ER fluids. Studies have demonstrated the successful ER performances of silicon-containing oxide materials: Tian et al. reported a zeolite-based ER fluid with a high dielectric constant [36]; Ma et al. synthesized an anisotropic silicon-rubber-based ER elastomer with 30 vol% TiO_2_-urea core-shell materials in silicone oil [37]; Méheust et. al. reported the ER activity of synthetic clay/mica-based ER fluids [38].

Mica consists of layered silicate materials. It naturally occurs in the earth as a mineral, and various methods have been developed for fabricating synthetic mica (mica) [39]. Natural and synthetic mica materials are utilized in various industrial fields owing to their benign characteristics, including nontoxicity, chemical stability, transparency, and glossiness [40,41]. Therefore, mica materials are widely employed in cosmetics, coating systems, and printing inks [42,43,44]. Furthermore, metal oxides, such as TiO_2_ and SiO_2_, can be coated onto the surface of mica to induce the light interference effect and consequently obtain various color-expressing pearlescent materials [45]. The light interference effects are facilitated on the surface of the metal oxide layer owing to the differences in refractive indices of the mica and the coated materials [46,47]. In addition, the thickness of the metal oxide coating layer can be manipulated to alter the color of pearlescent pigments.

While TiO_2_, silicon-containing oxides, and mica have been previously used as ER materials, research thus far has mainly focused on altering the shape, size, and porosity of materials [48]. In addition, numerous ER studies have demonstrated the incorporation or mixing of various materials, but the colors of ER fluids are confined to white, gray, and transparent. Therefore, negligible attention has been devoted to the synthesis of ER fluids of various colors. As the colors of conventional ER fluids are indistinguishable, confirming the performance of ER fluids with the naked eye and selecting an appropriate fluid for application are challenging. Therefore, developing ER fluids that can be selected for an application by correlating their color and performance is essential.

Herein, we developed six novel differently colored ER fluids by synthesizing various TiO_2_-coated colored mica (color mica/TiO_2_) materials. To the best of our knowledge, this is the first study to demonstrate the performances of ER fluids of various colors and their underlying mechanisms. The color mica/TiO_2_-based ER fluids were synthesized by coating TiO_2_ on synthetic base materials; the thickness of the TiO_2_ layer was manipulated to obtain a different color for each fluid—white, yellow, red, violet, blue, and green—owing to differences in the light interference effects. Significantly, the ER performance of each color mica/TiO_2_-based ER fluid changed according to the thickness of the TiO_2_ layer. Specifically, among the synthesized samples, the white mica/TiO_2_-based ER fluid demonstrated the highest ER activity (95.1 Pa), whereas green mica/TiO_2_-based ER fluids exhibited the lowest (51.1 Pa). Therefore, it was possible to select an appropriate ER fluid for a required application by simply evaluating the color. Additionally, the underlying ER mechanism for each color mica/TiO_2_-based ER fluid was revealed. The ER performance was directly affected by the amounts of Na^+^ and Ca^2+^ ions present in the ER fluids. In addition, the dispersion stability of each colored ER fluid was different, and the stability increased with decreasing TiO_2_ thickness. The synergistic contributions of ion polarizability and dispersion stability affect the ER performance of each ER fluid. In conclusion, we expect that these novel colored ER fluids will pave the way for the development of future smart stress control systems by allowing the facile selection of the appropriate ER fluid for an application by evaluating its color.

## 2. Materials and Methods

### 2.1. Materials

Mica (powder type) was purchased from CQV Co., Ltd. (Jincheon, Korea). Titanium tetrachloride (99.9%) and silicone oil (viscosity = 100 cSt) were purchased from Sigma-Aldrich (Burlington, MA, USA). Sodium hydroxide (40.0%) and ethanol (EtOH, 99.9%) were purchased from Samchun Chemical Company (Seoul, Korea). All chemicals were used as received without any additional purification.

### 2.2. Synthesis of Color-Mica/TiO_2_ Materials

Various color mica/TiO_2_ materials were synthesized using a typical sol–gel method. Mica (10.5 g) was dispersed in deionized water (100 mL) under vigorous magnetic stirring for 1 h. A constant heat of 90 °C was then applied to the solution, and TiCl_4_ (2.0 mL) was continuously added dropwise with constant stirring—the amount of TiCl_4_ added determined the color of the material. The sol–gel reaction proceeded for 4 h, during which NaOH (20.0%) was slowly added to the suspension to maintain the pH value at 1.8; pH preservation is vital for the successful coating of the TiO_2_ layer on mica. The resulting color mica/TiO_2_ materials were collected via vacuum filtration, washed several times with water to remove residues, and dried in an oven overnight at 90 °C. The final dry materials obtained were: white mica/TiO_2_, yellow mica/TiO_2_, red mica/TiO_2_, violet mica/TiO_2_, blue mica/TiO_2_, and green mica/TiO_2_. Figure 1 shows a schematic of the synthesis process.

### 2.3. Characterization

The morphological structures of the mica and color mica/TiO_2_ materials were investigated via field emission scanning electron microscopy (FE-SEM, S-4800, Hitachi, Tokyo, Japan); cross-sectional samples of the color mica/TiO_2_ materials were prepared for the analysis using a focused ion beam (FIB, LYRA3 GMH, TESCAN, Brno, The Czech Republic) system. The elemental compositions (Si and Ti) of the color mica/TiO_2_ materials were obtained using an energy-dispersive X-ray spectroscopy (EDS) system (EX-250, HORIBA, Ltd., Kyoto, Japan) integrated with the FE-SEM instrument. The molecular structures of the materials were analyzed using a Fourier-transform infrared (FT-IR) instrument (Nicolet iS10, Thermo Fisher Scientific, Waltham, MA, USA). The concentrations of Na^+^ and Ca^2+^ ions in the materials were determined using an ion chromatography system (930 Compact IC Flex, Metrohm, Herisau, Switzerland). The electrical conductivities of the materials in the pellet form were obtained using a two-point probe system (MCP-HT450, Mitsubishi, Tokyo, Japan).

### 2.4. Investigation of ER Properties

To prepare the color mica/TiO_2_-based ER fluids (3.0 wt%), the dried ER materials (0.3 g) were ultrasonically dispersed in silicone oil (11 mL, viscosity = 100 cST), followed by vigorous magnetic stirring overnight. No additional additives were added to the prepared ER fluids. The ER investigation of the samples was performed using a rheometer (MCR 302, Anton Parr, Graz, Austria) with a cup, concentric cylinder conical geometry, and high-voltage generator (HCN 7E-12500, Fug Elektronik, Schechen, Germany). The gap between the cup and the geometry was set to 1.00 mm on each side without any interference. The ER measurements were initiated by adding well-dispersed ER fluids into the cup along with a rotor. A mechanical shear rate of 10.0 s^−1^ was applied for 5 min to achieve stability and homogeneous placement of the fluid in the cup. Finally, the *E*-field was applied to analyze the ER performances of the different ER fluids under various conditions.

## 3. Results and Discussion

### 3.1. Structure and Morphology of Color-Mica/TiO_2_ Materials

To visualize the color appearance, the color mica/TiO_2_ materials were dissolved in EtOH and observed by the naked eye. Clearly, each color mica/TiO_2_ material expressed a distinct color (Figure 2a). The color change of a color mica/TiO_2_ material can be explained as follows: When incident light penetrates the color mica/TiO_2_ material, a certain amount of light is reflected from the nica–TiO_2_ interface. This reflected light encounters new incident light, facilitating the light interference effect. Varying the thickness of the TiO_2_ layer changes the traveling distances of the incident light, causing different interference effects involving various wavelengths, thereby producing different colors. This correlation between the ER material color and TiO_2_ layer thickness is shown in Figure 2b. The refractive indices of mica and TiO_2_ are 1.6 and 2.5, respectively [49,50]. According to Fresnel’s law of reflection, the larger the difference in the refractive index between two materials, the higher the light reflectance at the interface of the materials [51,52].

The morphologies of the color mica/TiO_2_ materials were investigated using FE-SEM (Figure 3). The particle size of the color mica/TiO_2_ materials was determined to be approximately 25 μm. Significantly, a clear separation between the particles was observed for all the color mica/TiO_2_ materials, which is in contrast with the aggregated mica particles; this particle separation occurred owing to the presence of the TiO_2_ layer. Notably, the surfaces of mica and color mica/TiO_2_ materials were completely different: compared to the smooth surface of mica, the color mica/TiO_2_ materials displayed rough surfaces owing to the TiO_2_ coating. In addition, the size of the TiO_2_ particles increased with increasing TiO_2_ thickness.

To further investigate the layer of each color mica/TiO2 material, FIB analysis was conducted on cross-sectional samples of mica and the color mica/TiO_2_ materials, and the thickness of the TiO_2_ layers was determined (Figure 4). For the mica sample, the measured thickness was approximately 728.6 nm. For the color mica/TiO_2_ materials, the thicknesses of the TiO_2_ layers of the white mica/TiO_2_, yellow mica/TiO_2_, red mica/TiO_2_, violet mica/TiO_2_, blue mica/TiO_2_, and green mica/TiO_2_ materials were approximately 52.3, 84.4, 115.7, 138.8, 164.9, and 187.5 nm, respectively. Clearly, TiO_2_ layers were successfully introduced onto the mica in the color mica/TiO_2_ materials, and the color of each material changed according to the different light interference effects of the respective TiO_2_ layers.

EDS analysis was conducted to examine the elemental compositions (Si and Ti) of mica and various color mica/TiO_2_ materials. For mica, only Si was detected, and no Ti was detected. In contrast, Ti was detected in all the color mica/TiO_2_ materials. The Ti proportions for white mica/TiO_2_, yellow mica/TiO_2_, red mica/TiO_2_, violet mica/TiO_2_, blue mica/TiO_2_, and green mica/TiO_2_ were 21.0%, 31.5%, 41.0%, 49.2%, 55.7%, and 61.9%, respectively. The total Ti composition of each color mica/TiO_2_ material increased with increasing TiO_2_ thickness, which is in accordance with the cross-sectional FE-SEM observation. The detailed elemental compositions of mica and the color mica/TiO_2_ materials are listed in Table 1.

FT-IR analysis was conducted to investigate the molecular structures of mica and the color mica/TiO_2_ materials, as shown in Figure 5. In the case of mica, characteristic peaks were observed at 970, 800, and 695 cm^−1^, which correspond to silicon-containing oxides with asymmetric Si–O–Si stretching vibrations, Si–O symmetric stretching vibrations, and Si–O symmetric bending, respectively [53,54,55]. Significantly, the peaks characteristic to mica diminished in the absorbance curves of all the color mica/TiO_2_ materials, indicating the successful coating of TiO_2_. Moreover, the absorbance increased after 830 cm^−1^ for the color mica/TiO_2_ materials owing to the presence of the TiO_2_ layers, which absorbs in the near-infrared (NIR) region.

### 3.2. Suitability Tests of Color-Mica/TiO_2_ Materials for ER Application

The suitability of various color mica/TiO_2_ materials for ER applications was investigated by determining their ion concentration, electrical conductivity, and dispersion stability. First, ion chromatography was used to identify the ion concentrations of the color mica/TiO_2_ materials. Ion chromatography revealed the presence of Na^+^ and Ca^2+^ in all the color mica/TiO_2_ materials. Notably, the concentrations of both Na^+^ and Ca^2+^ ions decreased as the thickness of the TiO_2_ layer increased, indicating that the ion species originated from the mica material, which contains various minerals. Detailed concentrations of Na^+^ and Ca^2+^ ions in various color mica/TiO_2_ materials are listed in Table 2. Previous studies have demonstrated that ion species enhance the ER activity via ion polarization effects [17]. Therefore, the ER performance of color mica/TiO_2_ may be affected by the ion concentration, and a high concentration may have a benign effect on the ER performance.

The electrical conductivities of various color mica/TiO_2_ materials were investigated to examine their suitability for ER applications, as electrical conductivity is a key factor that affects the ER performance of a material. Previous studies have demonstrated that the electrical conductivity should be near ~10^−9^ S m^−1^ for achieving the ideal ER performance of a material [56]. Moreover, the electrical conductivity mismatch between the materials and dispersing medium can facilitate the ER effect [25,57]. The electrical conductivities for white mica/TiO_2_, yellow mica/TiO_2_, red mica/TiO_2_, violet mica/TiO_2_, blue mica/TiO_2_, and green mica/TiO_2_ were determined to be 2.4 × 10^−9^, 2.1 × 10^−9^, 1.7 × 10^−9^, 1.4 × 10^−9^, 1.1 × 10^−9^, and 0.9 × 10^−9^ S m^−1^, respectively; the conductivity of silicone oil is ~1 × 10^−13^ S m^−1^. Therefore, the electrical conductivities of various color mica/TiO_2_ materials were in a suitable range for ER applications, and the electrical conductivity of each material was at least two orders of magnitude higher than that of silicone oil, thereby consequently promoting the ER effect of the materials [25].

Finally, the dispersion stabilities of the color mica/TiO_2_ materials were examined (Figure 6). For ER applications, dispersion stability is another key factor affecting ER performance, as the sedimentation of materials from the media may disrupt the formation of rigid fibril-like structures. To examine the dispersion stability, each color mica/TiO_2_ material was thoroughly mixed with silicone oil at a concentration of 3.0 wt%. Each fluid was well-dispersed in the original state, but the dispersed materials gradually sedimented over time, reaching the equilibrium state at 20 h. After 20 h, each color mica/TiO_2_-based ER fluid exhibited favorable dispersion stability for ER application. The dispersion stability of the materials decreased in the order of white mica/TiO_2_ > yellow mica/TiO_2_ > red mica/TiO_2_ > violet mica/TiO_2_ > blue mica/TiO_2_ > green mica/TiO_2_, indicating that the dispersion stability of the material decreased with increasing TiO_2_ content.

### 3.3. ER Performances of Color-Mica/TiO_2_-Based ER Fluids

The ER characteristics of the six differently colored ER fluids were examined by determining the shear stress under various conditions: as a function of shear rate, in *E* field on–off conditions, and at various *E* field strengths (Figure 7). First, ER activities were measured as a function of shear rate when an *E*-field strength of 3.0 kV mm^−1^ was applied (Figure 7a). All the ER fluids experienced immediate shear stress upon the application of the *E* field, owing to the electrostatic forces between the dispersed materials in the dispersing media [58]; the shear stress resulted in the formation of fibril-like structures. In the low shear rate region, all ER fluids exhibited Bingham plastic behavior, which was ascribed to the electrostatic forces between materials that were stronger than the hydrodynamic forces generated by mechanical shear [59]. Past the critical shear rate (τ_c_), Newtonian behavior was exhibited as the shear stress increased proportionally with shear rate, indicating the dominance of hydrodynamic forces over electrostatic forces [60]. The determined ER activities for the white, yellow, red, violet, blue, and green mica/TiO_2_-based ER fluids were 95.1, 81.9, 71.2, 63.3, 57.5, and 51.1 Pa, respectively; the ER performance of the white mica/TiO_2_-based ER fluid was the highest among the various color mica/TiO_2_-based ER fluids. This trend mirrors that of the ER performance, which was in the order of white > yellow > red > violet > blue > green materials. These trends in relation to ER activity are in accordance with the experimental results of various suitability tests. As discussed earlier, the white mica/TiO_2_ material displayed the highest ion concentrations, the largest electrical conductivity difference with the dispersing media, and excellent dispersion stability compared to the other color mica/TiO_2_ materials. In contrast, the green mica/TiO_2_ material demonstrated low ion concentrations, electrical conductivity, and dispersion stability, thereby exhibiting the poorest ER performance. The tentative mechanism for the different ER performances of white and green mica/TiO_2_-based ER fluids is shown in Figure 7b.

*E*-field on–off tests were performed to examine the reversibility of color mica/TiO_2_-based ER fluids (Figure 7c). Without an applied *E* field, all ER fluids were in the rest position, experiencing a shear stress of less than 1.0 Pa. However, when an *E* field of 3.0 kV mm^−1^ was applied, shear stresses were observed immediately for all the ER fluids. Similarly to previous ER measurements, the magnitude of shear stresses experienced by each fluid followed the order of white > yellow > red > violet > blue > green. When the applied *E* field was turned off, the shear stresses immediately returned to their initial states. Therefore, all the color mica/TiO_2_-based ER fluids clearly exhibited reversible ER features. Finally, dynamic yield stresses of various color mica/TiO_2_-based ER fluids were measured as a function of *E* field strength at a fixed shear rate of 0.1 s^−1^ (Figure 7d). The applied *E* field was then elevated from 1.0 to 4.0 kV mm^−1^ in increments of 1.0 kV mm^−1^. The dynamic yield stress of all the ER fluids increased by approximately 1.5 times the power of the *E* field strength in the region. All the ER fluids exhibited stable dynamic yield stresses without any electrical shorts.

To examine the real-time response of each ER fluid, optical microscopy (OM) was performed (Figure 8). The well-dispersed color mica/TiO_2_-based ER fluids were placed between two electrodes. When an *E* field of 1.0 kV mm^−1^ was applied, the dispersed materials formed fibril-like structures within a few tens of milliseconds. Notably, each structure exhibited the complementary color of the corresponding color mica/TiO_2_ material; specifically, the structures in the white, yellow, red, violet, blue, and green mica/TiO_2_-based ER fluids displayed complementary colors of black, blue, green, yellowish-green, yellow, and red, respectively. Therefore, the OM analysis proved the rapid response time of the ER fluids, while also clearly evidencing the successful synthesis of the color mica/TiO_2_-based ER fluids with their distinct colors.

## 4. Conclusions

In conclusion, Mica/TiO_2_ materials with various colors—white, yellow, red, violet, blue, and green—were successfully prepared by the facile coating of a TiO_2_ layer on mica material using a sol–gel method, utilizing TiCl_4_ as a precursor. To control the color of each material, the amount of TiCl_4_ added was adjusted during the reaction to vary the thickness of the TiO_2_ layer, and consequently vary the light interference effects. ER suitability tests, including ion concentration, electrical conductivity, and dispersion stability measurements, were conducted for each color mica/TiO_2_ material. The results obtained for all the color mica/TiO_2_ materials were in suitable ranges for use in ER applications. The ER activities of the color mica/TiO_2_-based ER fluids were measured using various ER testing methods. Notably, with increasing thickness of the TiO_2_ layer, the ER performance decreased: the white mica/TiO_2_-based ER fluid exhibited the highest ER performance (95.1 Pa) owing to its high ion concentration and dispersion stability, and the green mica/TiO_2_-based ER fluid demonstrated the lowest performance (51.1 Pa). Hence, our novel colored ER fluids showed a correlation of color with ER performance, which may establish a new standard for selecting ER fluids by visually evaluating the colors.

## Figures and Tables

**Figure 1 nanomaterials-12-03113-f001:**
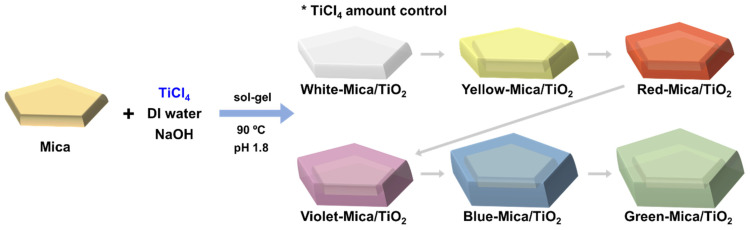
Schematic of the synthesis of color mica/TiO_2_ materials via the sol–gel method.

**Figure 2 nanomaterials-12-03113-f002:**
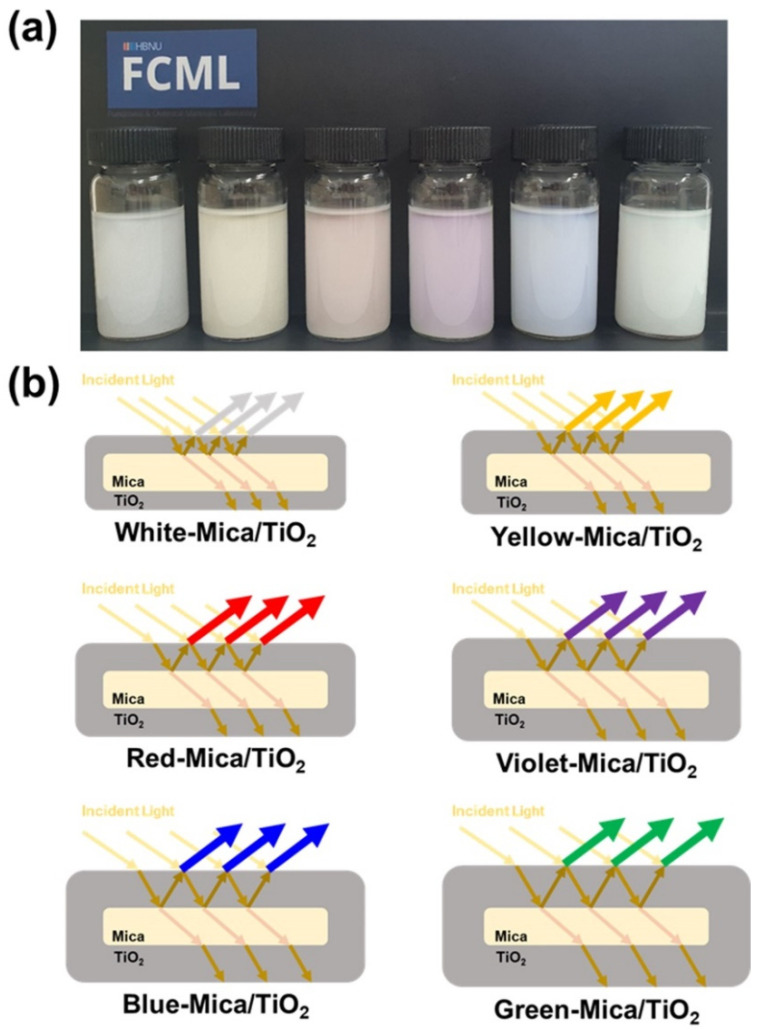
(**a**) Digital photographs of white mica/TiO_2_, yellow mica/TiO_2_, red mica/TiO_2_, violet mica/TiO_2_, blue mica/TiO_2_, and green mica/TiO_2_ materials (from left to right) dispersed in EtOH. (**b**) Schematic for the underlying mechanism of the color mica/TiO_2_ materials with different colors.

**Figure 3 nanomaterials-12-03113-f003:**
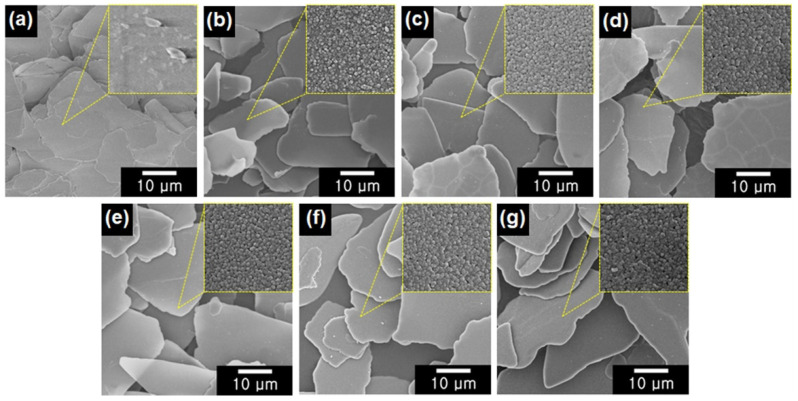
FE-SEM micrographs of (**a**) mica, (**b**) white mica/TiO_2_, (**c**) yellow mica/TiO_2_, (**d**) red mica/TiO_2_, (**e**) violet mica/TiO_2_, (**f**) blue mica/TiO_2_, and (**g**) green mica/TiO_2_ (inset: FE-SEM images of the surface of each material).

**Figure 4 nanomaterials-12-03113-f004:**
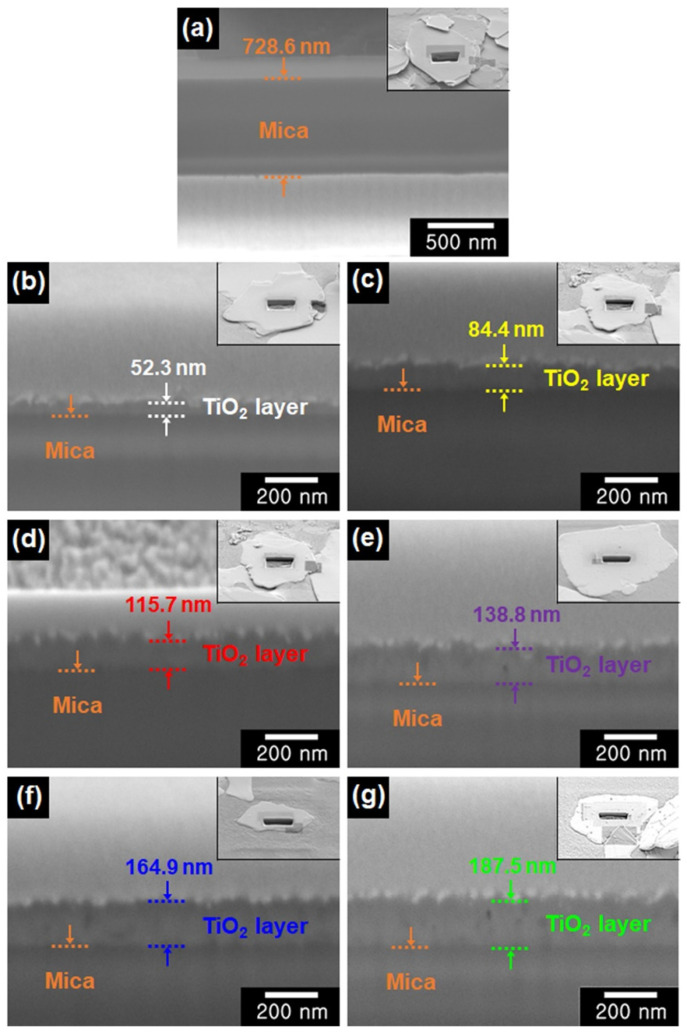
Cross-sectional FE-SEM images of (**a**) mica, (**b**) white mica/TiO_2_, (**c**) yellow mica/TiO_2_, (**d**) red mica/TiO_2_, (**e**) violet mica/TiO_2_, (**f**) blue mica/TiO_2_, and (**g**) green mica/TiO_2_ prepared using a focused ion beam (FIB) system. Regions of Mica are marked as dotted orange lines (inset: low magnification FE-SEM images of cross-sectional samples).

**Figure 5 nanomaterials-12-03113-f005:**
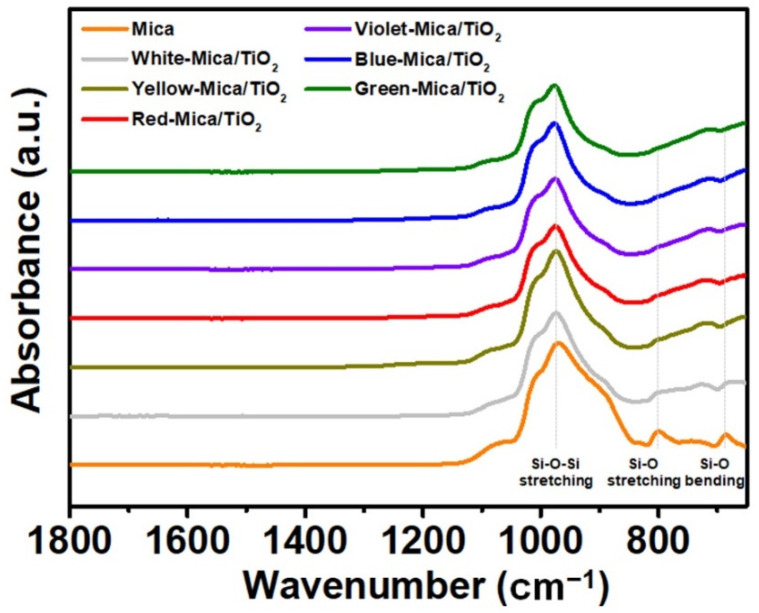
FT-IR spectra of mica and color mica/TiO_2_ materials.

**Figure 6 nanomaterials-12-03113-f006:**
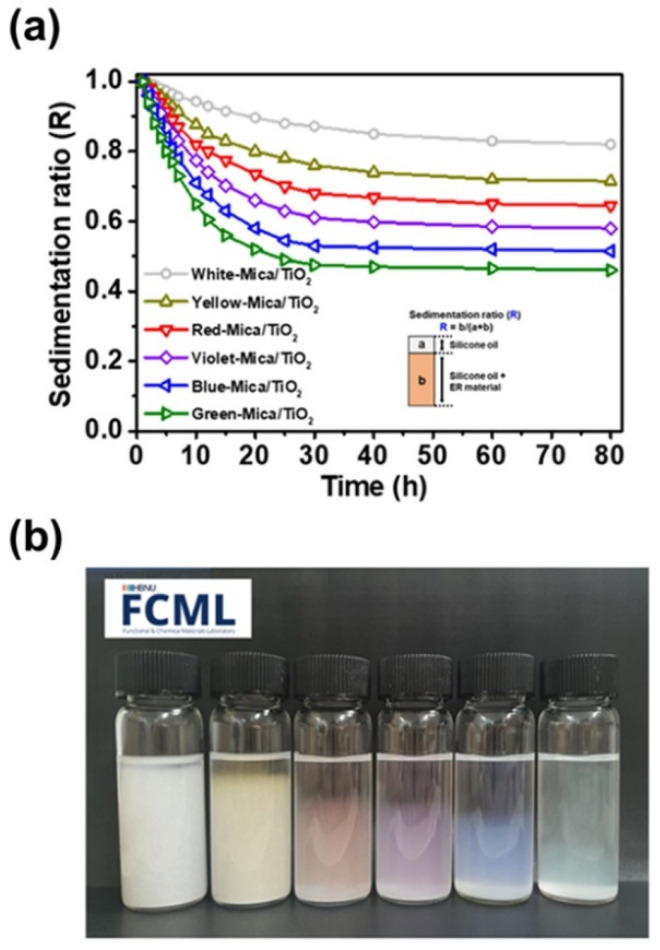
(**a**) Dispersion stabilities of color mica/TiO_2_-based ER fluids dispersed in silicone oil (3.0 wt%) and (**b**) digital images of color mica/TiO_2_-based ER fluids after 20 h.

**Figure 7 nanomaterials-12-03113-f007:**
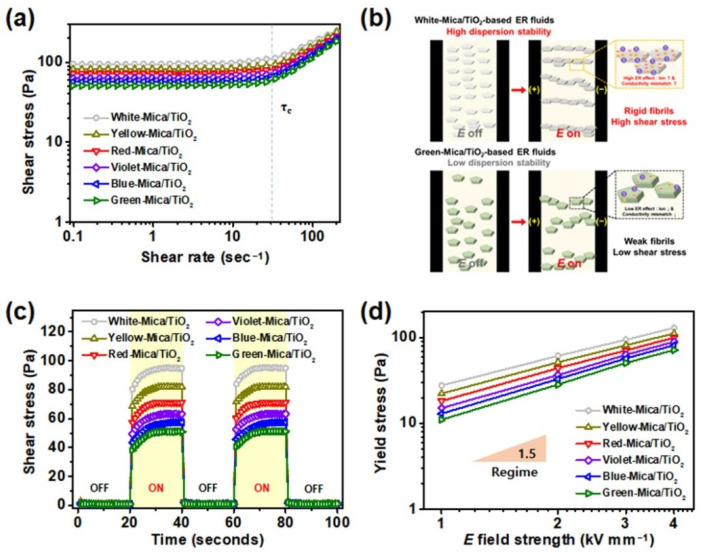
(**a**) Shear stresses of various color mica/TiO_2_-based ER fluids (3.0 wt%) as a function of shear rate with an applied *E* field strength of 3.0 kV mm^−1^. (**b**) Tentative mechanism for high and low ER efficiency difference between white and green mica/TiO_2_-based ER fluids according to the ion concentration, electrical conductivity, and dispersion stability. (**c**) *E*-field on–off test results with a fixed shear rate of 0.1 s^−1^ and (**d**) yield stresses of various color mica/TiO_2_-based ER fluids (3.0 wt%) as a function of the *E* field strength with a fixed shear rate of 0.1 s^−1^.

**Figure 8 nanomaterials-12-03113-f008:**
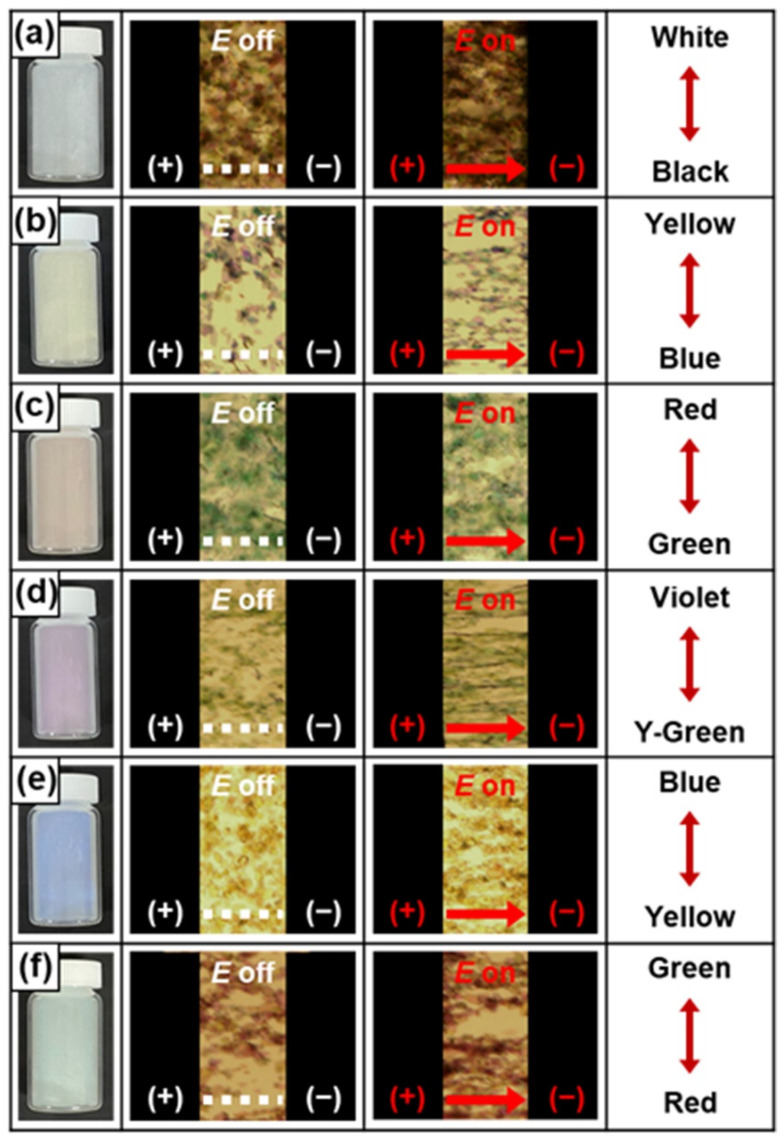
Visual and OM images of fibril-like structures of (**a**) white, (**b**) yellow, (**c**) red, (**d**) violet, (**e**) blue, and (**f**) green mica/TiO_2_-based ER fluids (3.0 wt% in silicone oil) when an *E* field of 3.0 kV mm^−1^ was applied. Complementary colors were observed using an optical microscope.

**Table 1 nanomaterials-12-03113-t001:** Elemental compositions of mica and color mica/TiO_2_ materials *^a^*.

Samples	Elements (Atomic %)	
	Si	Ti
Mica	100.0	-
white mica/TiO_2_	79.0	21.0
yellow mica/TiO_2_	68.5	31.5
red mica/TiO_2_	59.0	41.0
violet mica/TiO_2_	50.8	49.2
blue mica/TiO_2_	44.3	55.7
green mica/TiO_2_	38.1	61.9

*^a^* Elemental compositions of samples was obtained using the EDS mode installed in the FE-SEM system, excluding all other elements except Si and Ti (beam current: 10.0 μA, accelerating voltage: 10.0 kV).

**Table 2 nanomaterials-12-03113-t002:** Concentrations of Na^+^ and Ca^2+^ ions in various color mica/TiO_2_ materials *^a^*.

Sample	Ion Concentration (ppm)	
	Na^+^	Ca^2+^
white mica/TiO_2_	21.8	13.2
yellow mica/TiO_2_	16.5	8.2
red mica/TiO_2_	12.1	5.8
violet mica/TiO_2_	11.0	2.7
blue mica/TiO_2_	9.0	1.3
green mica/TiO_2_	7.6	0.9

*^a^* Ion concentrations of samples were measured using a Metrohm 930 compact IC Flex system.

## Data Availability

Data are contained within the article.

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
