# Peer review of "Development of Novel Colorful Electrorheological Fluids"

_nanomaterials, 2022, doi:10.3390/nano12183113_

Round 1

Reviewer 1 Report

A series of TiO2 -coated synthetic mica materials with white, yellow, red, violet, blue, and green colors (referred to as Color-Mica/TiO2 materials) were fabricated via a facile sol-gel method. The ER performance of differently colored ER fluids was also affected by the electrical conductivity.

This pioneering study may provide a practical strategy for developing new ER fluid systems in future.

 Interesting direction. Of course, for an ER fluid, the color doesn't matter, but other interesting applications might be useful. You can publish as is!

Author Response

Response to Reviewer 1’s comments

Dear Reviewer

Thank you for reviewing our manuscript. We have substantially revised to reflect the reviewer’s comments on the manuscript. The following revisions were made on the basis of your comments.

---------------------------------------------------------------------------------------------------------

Reviewer 1

Comment) A series of TiO2 -coated synthetic mica materials with white, yellow, red, violet, blue, and green colors (referred to as Color-Mica/TiO2 materials) were fabricated via a facile sol-gel method. The ER performance of differently colored ER fluids was also affected by the electrical conductivity.

This pioneering study may provide a practical strategy for developing new ER fluid systems in future.

 Interesting direction. Of course, for an ER fluid, the color doesn't matter, but other interesting applications might be useful. You can publish as is!

Response:

Thank you for your reviewing and accepting our manuscript. As reviewer commented, we are submitting our manuscript for publication in Nanomaterials.

----------------------------------------------------------------------------------------------------------

We have made a considerable effort to address all the concerns of the referees in the revised manuscript.

We would like to ask your indulgence in considering our paper and highly welcome your valuable suggestions for its publication. Thank you for your consideration. I am looking forward to hearing from you sooner or later.

With all good wishes

Sincerely yours,

Prof. Chang-Min Yoon

Department of Chemical and

Biological Engineering,

Hanbat National University

Daejeon, Korea 34158

Tel) 82-42-821-1528

Fax) 82-42-821-1593

E-mail) [email protected]

Reviewer 2 Report

In Figure 4(g), the thickness of TiO2  is not clear, please clarify.

Author Response

Response to Reviewer 2’s comments

Dear Reviewer

Thank you for reviewing our manuscript. We have substantially revised to reflect the reviewer’s comments on the manuscript. The following revisions were made on the basis of your comments.

---------------------------------------------------------------------------------------------------------

Reviewer 2

Comment) In Figure 4(g), the thickness of TiO2 is not clear, please clarify.

Response:

Thank you for your helpful comment on this manuscript. As referee pointed out, we solely agreed on the necessity of clarifying the thickness of TiO2 layer in Figure 4(g). To account for the suggestion of referee, we have changed the font color of Figure 4(g) to brighter color for better clarification, as shown in Figure R1. In the revised manuscript, Figure R1 is inserted as Figure 4. Again, thank you for your helpful comment on our manuscript.

Figure R1. Cross-sectional FE-SEM images of (a) Mica, (b) White-Mica/TiO2, (c) Yellow-Mica/TiO2, (d) Red-Mica/TiO2, (e) Violet-Mica/TiO2, (f) Blue-Mica/TiO2, and (g) Green-Mica/TiO2 prepared using a focused ion beam (FIB) system. Regions of Mica are marked as dotted orange lines [inset: low magnification FE-SEM images of cross-sectional samples].

----------------------------------------------------------------------------------------------------------

We have made a considerable effort to address all the concerns of the referees in the revised manuscript.

We would like to ask your indulgence in considering our paper and highly welcome your valuable suggestions for its publication. Thank you for your consideration. I am looking forward to hearing from you sooner or later.

With all good wishes

Sincerely yours,

Prof. Chang-Min Yoon

Department of Chemical and

Biological Engineering,

Hanbat National University

Daejeon, Korea 34158

Tel) 82-42-821-1528

Fax) 82-42-821-1593

E-mail) [email protected]

Reviewer 3 Report

The paper can be accepted.

The ER response of some mica-based ER fluids has been tested correlating to their respective color.
This is a novelty that makes fast and easy the selection of ER fluids. The paper is well-written and easy to read. Experimental methods are detailed and clearly shown to allow for reproduction of this research. Results support the original idea that was proposed by authors. Sedimentation problem, being one of the most difficulties related to ER fluids applications, is tackled and well-characterized. The conclusions are consistent with the results.

Considering the interest for readers of Nanomaterials I recommend publication in present form.

Author Response

Response to Reviewer 3’s comments

Dear Reviewer

Thank you for reviewing our manuscript. We have substantially revised to reflect the reviewer’s comments on the manuscript. The following revisions were made on the basis of your comments.

---------------------------------------------------------------------------------------------------------

Reviewer 3

Comment) The paper can be accepted. The ER response of some mica-based ER fluids has been tested correlating to their respective color. This is a novelty that makes fast and easy the selection of ER fluids. The paper is well-written and easy to read. Experimental methods are detailed and clearly shown to allow for reproduction of this research. Results support the original idea that was proposed by authors. Sedimentation problem, being one of the most difficulties related to ER fluids applications, is tackled and well-characterized. The conclusions are consistent with the results. Considering the interest for readers of Nanomaterials I recommend publication in present form.

Response:

Thank you for your reviewing and accepting our manuscript. As reviewer commented, we are submitting our manuscript for publication in Nanomaterials.

----------------------------------------------------------------------------------------------------------

We have made a considerable effort to address all the concerns of the referees in the revised manuscript.

We would like to ask your indulgence in considering our paper and highly welcome your valuable suggestions for its publication. Thank you for your consideration. I am looking forward to hearing from you sooner or later.

With all good wishes

Sincerely yours,

Prof. Chang-Min Yoon

Department of Chemical and

Biological Engineering,

Hanbat National University

Daejeon, Korea 34158

Tel) 82-42-821-1528

Fax) 82-42-821-1593

E-mail) [email protected]
